# An Adversarial Attack Method against Specified Objects Based on Instance Segmentation

**Dapeng Lang [1,2,\*]**, **Deyun Chen [1,\*]**, **Sizhao Li [3]** and **Yongjun He [1]**

1   School of Computer Science and Technology, Harbin University of Science and Technology,
    Harbin 150080, China
2   College of Computer Science and Technology, Harbin Engineering University, Harbin 150001, China
3   Faculty of Engineering, The Chinese University of Hong Kong, Hong Kong, China
*   Correspondence: langdapeng@outlook.com (D.L.); chendeyun@hrbust.edu.cn (D.C.)

**Abstract:** The deep model is widely used and has been demonstrated to have more hidden security risks. An adversarial attack can bypass the traditional means of defense. By modifying the input data, the attack on the deep model is realized, and it is imperceptible to humans. The existing adversarial example generation methods mainly attack the whole image. The optimization iterative direction is easy to predict, and the attack flexibility is low. For more complex scenarios, this paper proposes an edge-restricted adversarial example generation algorithm (Re-AEG) based on semantic segmentation. The algorithm can attack one or more specific objects in the image so that the detector cannot detect the objects. First, the algorithm automatically locates the attack objects according to the application requirements. Through the semantic segmentation algorithm, the attacked object is separated and the mask matrix for the object is generated. The algorithm proposed in this paper can attack the object in the region, converge quickly and successfully deceive the deep detection model. The algorithm only hides some sensitive objects in the image, rather than completely invalidating the detection model and causing reported errors, so it has higher concealment than the previous adversarial example generation algorithms. In this paper, a comparative experiment is carried out on ImageNet and coco2017 datasets, and the attack success rate is higher than 92%.

**Keywords:** adversarial examples; adversarial patch; instance segmentation; neural networks; deep model security

## 1. Introduction

Deep neural network surpasses the previous machine learning models in many aspects. Due to its strong expression ability, the deep model has achieved great success in the fields of speech recognition [1], computer vision [2], network security [3] and so on. In a large number of application scenarios based on deep learning, such as decision-making, finance and automatic driving [4], many application environments have very high requirements for security [5]. Therefore, the robustness of a deep neural network is particularly important.

The concept of "adversarial examples" was first introduced by Szegedy et al. [2], who used perturbation on the pixels of the original pixels to induce significant accuracy reduction. Nguyen et al. [6] proposed that deep learning models can classify images with high confidence even when they are not recognized by humans. In 2016, Pedro et al. [7] proposed that the application of adversarial examples occupies great space in images. Since then, deep networks and adversarial examples have become a research boom. Goodfellow et al. [8] explained that the high-dimensional linearity of deep networks is the principle of adversarial example generation, and iterative algorithms around the generation of adversarial examples started to emerge.

The mainstream adversarial example generation algorithms include the following categories. Novel attack methods are mostly evolved based on these algorithms. The

FGSM (fast gradient sign method) algorithm proposed by Goodfellow et al. [8] has become one of the most fundamental white box adversarial example algorithms with different tasks. Another popular white box algorithm is PGD (projected gradient descent) [9]. To ensure that the solution remains in the constrained set when solving a constrained convex optimization problem, the algorithm uses the values projected to the constrained set for the gradient descent update parameters. Other classical attack methods include DeepFool [10], universal adversarial perturbations (UAP) [11] and Carlini and Wagner attacks (C & W) [12]. In addition, there are methods that generate adversarial examples directly by data augmentation [13] or attack other operations in the deep learning image processing pipeline (e.g., image resizing) [14].

Worryingly, the existence of adversarial examples is not due to the weak robustness of a deep model or bugs caused by programmers but due to the structure of the deep model itself. The inexplicability of deep models generates some visual features that violate human intuition [15].

The adversarial attack is very different from the traditional network attack and is more hidden and more difficult to detect and defend against [16]. It is not only to threaten the deep model but also to expand the attack range of adversarial attacks [4]. It is common in the deep learning model, which shows that the deep neural network is very fragile and shows weak robustness [17].

However, the current adversarial algorithms have theoretically demonstrated the robustness problem of deep models, but they still face considerable problems in practice. In the process of a military UAV attack, directional hiding is carried out for a certain type of specified object, and the state-of-the-art detector is deceived by adversarial examples. In the literature [18], an image-based attention mechanism is proposed to locate the key optimization region against perturbations. The foremost strongest first-order attack so far is presented in the Ref. [9], which is used to test the robustness of deep models; in specific complex application scenarios, attackers may launch an attack by generating adversarial patches starting with random patterns on the glasses [19] or [20] clothing.

The existing algorithms attack all objects in the image, which is easy to cause vigilance of the defender. Furthermore, they are not sensitive to attack objects, so they may have poor scalability and are challenging to be popularized in the targeted fixed-point attack field. The algorithm proposed in this paper has been improved in the above aspects.

## 2. Contributions of This Paper

In this paper, we propose an adversarial examples generation algorithm based on instance segmentation. The algorithm separates the objects based on scenario requirements and hides the specified object by restricting the generation area of adversarial examples. The generated adversarial examples are verified against the state-of-the-art YOLOv4 detector [21]. The algorithm not only can take into account the advantages of high concealment but also the fast convergence. The algorithm flow is shown in Figure 1.

As shown in Figure 1, the traditional method of adversarial attacks is mainly to generate adversarial examples for the whole image through the gradient descent method on a clean image. The adversarial perturbation visible to humans is generated for the whole image in the upper part of Figure 1d. It realizes the deception of the detector, and the detector cannot detect three vehicles in the image. Two of them are complete and the third is only part. The lower part of Figure 1 is the algorithm flow proposed in this paper. We extract the target object outline from the image through pre-extraction semantic analysis (Figure 1b). Adversarial examples are generated in the area delineated by the edge of the mask. As shown in the lower part on the right side of Figure 1d, the detector turns a blind eye to object number two but can successfully detect other objects.

A more typical attack scenario is shown in Figure 2, which includes several different objects, including three pedestrians, two cars and a truck. The algorithm launches an attack on one of the objects in the image. The effect of the attack is shown in Figure 2.

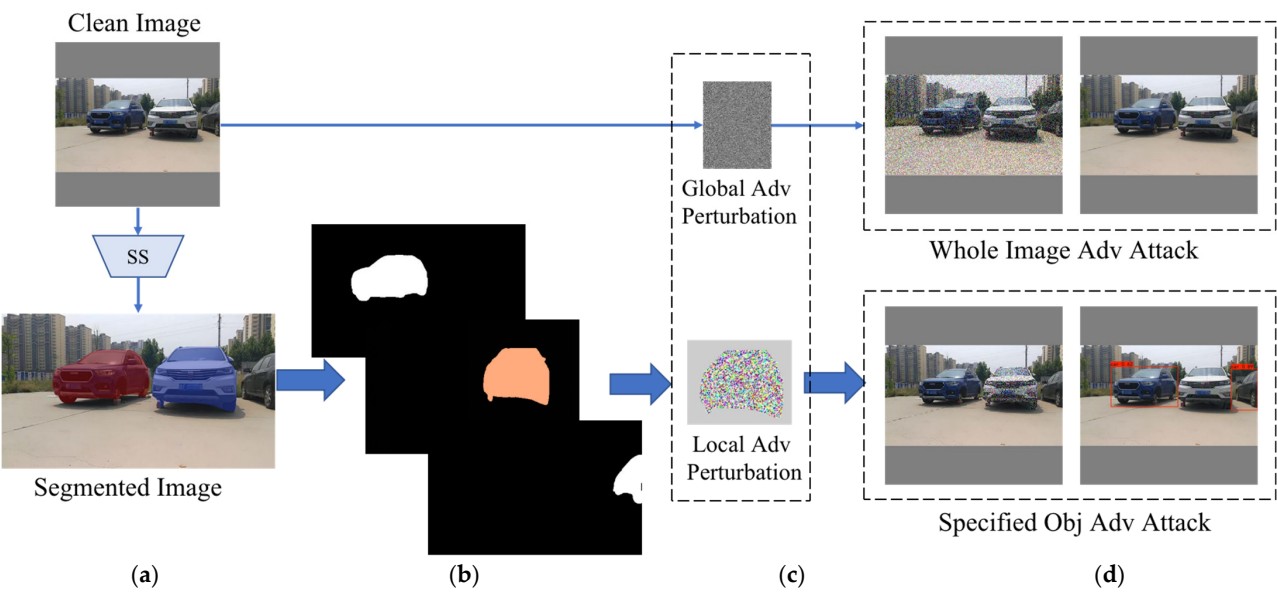

**Figure 1.** Comparison of different attack method processes. (**a**) Image Segmentation. (**b**) Mask Generation. (**c**) Adversarial Perturbation Generation. (**d**) Adversarial Example Generation. The upper half of the figure shows the traditional adversarial example generation algorithm. The bottom counterpart is the algorithm proposed in this paper.

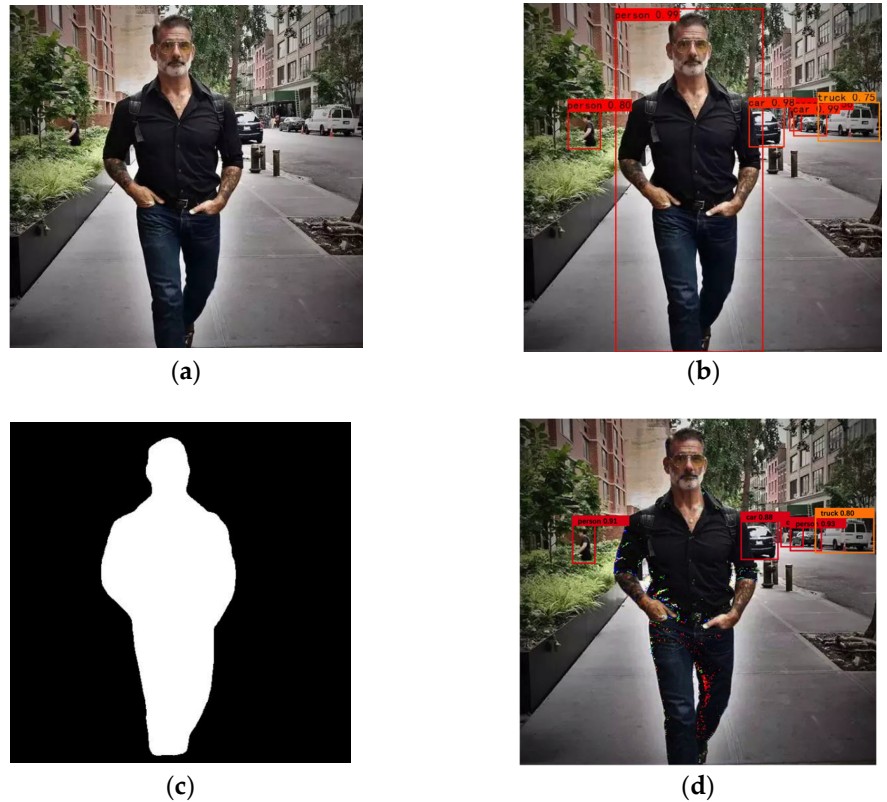

**Figure 2.** Adversarial examples can successfully deceive detectors: (**a**) is a clean picture; (**b**) shows that the detector can successfully detect all objects in the image when the image is not under attack; (**c**) is the generation of the mask matrix of the attacked object based on the segmentation algorithm; (**d**) shows the attack results. The detector cannot detect the specific object but can correctly detect other objects.

Experiments show that the algorithm proposed in this paper can efficiently generate adversarial examples and is suitable for multiple complex application scenarios. This paper makes the following contributions:

- Different from existing methods, which are based on full images of adversarial examples, our algorithm can attack local objects in the image according to the application scenario requirements and converges faster and is more flexible.
- Based on the optimized framework in this paper, two different forms of attack are implemented in this paper, namely adversarial perturbation and adversarial patching. Both of them can produce a good attack effect on the objects within the area.
- The proposed attack method can delineate one or more attack areas according to the requirements of the complex scenes and attack the objects in the areas, making them undetectable to the detector. Even partially exposed objects can be hidden.
- Compared with several classic adversarial algorithms, our algorithm has the advantages of fast convergence and less modified pixels. We have implemented multiple experiments on ImageNet and coco2017 datasets. The performance of the algorithm exceeds the traditional attack methods.

This paper verifies the transferability of the algorithm between multiple detectors through the transferability test. The algorithm can generate adversarial examples with high confidence.

## 3. Related Works

### 3.1. Adversarial Examples Generation

It is found in Ref. [1] that, even for noise without any meaning sensed by humans, the deep model may still provide classification results with high confidence. More and more evidence show that, due to the vulnerability of the structure of the deep model, a noisy image with different degrees of perturbation may also deceive classification systems.

With more and more extensive applications and in-depth research of adversarial examples, the related generation principle also deepens people's understanding of the robustness and interpretability of the deep model [2]. Goodfellow et al. [1] discussed the basic principles of adversarial examples. Many new attack methods and example generation methods are developed on the basis of this theory. The increasing aggressiveness of adversarial attacks, in turn, promotes the defense ability of the deep model against attacks. In fields where deep learning is widely used, such as video surveillance, dynamic tracking, medical finance and other fields, the adversarial examples have been able to successfully attack the real application system, which has attracted significant attention of security personnel and researchers from both red and blue sides. Because the existence of adversarial attacks is caused by the inherent structure of the deep model, excellent concealment and the effect of the attack can be realized without modifying too much information.

Common classic attack algorithms include L-BFGS [2], FGSM [1], DeepFool [15], PGD [9], etc. Inspired by these algorithms, new algorithms have sprung up one after another. For example, with the attack method of saliency mapping based on Jacobian [20], its core idea is to find the pixels that need to be modified in the image so as to minimize the pixels that need to be modified in the image. Corresponding to digital attacks, the physical attack needs to print (or project) the adversarial examples through printers, projectors and other devices to maintain the attack effect in the real world.

In Ref. [22] by Kurakin et al., the physical adversarial examples can mislead the detector even if the size, angle and position are adjusted accordingly after printing. Sharif et al. [23] proposed a new type of glasses with an adversarial effect. The glasses are printed with unique patterns so that the face detection system can recognize the wearer as others. This type of attack can mislead the classifier and misjudge the attacked objects as the classification specified by the attacker.

### 3.2. Adversarial Patches Generation

The work of adversarial examples focuses on attack examples that cannot be recognized by humans. The attack is highly hidden, but it is difficult to detect on the defensive side. Ref. [24] generates a visually obvious patch based on the image to be attacked. The author attached the patch to the image to mislead the detector. The application effect of the adversarial patch is more flexible than perturbation.

A physical adaptive patch is generated in Ref. [25]. The algorithm optimizes the optimization function composed of the non-printability score, the smooth score of the image and the objectness score, respectively. After the adversarial patch is printed, people can avoid the detection of a detector; Researchers [26] also consider the hidden attack on traffic signs in the automatic driving scene. Due to the addition of adversarial patches, the object detector based on a deep network cannot correctly identify its meaning. In this paper, a camouflage patch with a specified style is generated on the object image, which makes the patch look more natural and disguised; Ref. [27] shows a patch that can attack the object detector (YOLOv3 [21]) in the real world. Different from previous patches, this patch can detect and restrict almost all objects in the image without covering specified objects or even placing them in any position. The application range and accuracy of this method are greatly improved on the basis of Dpatch. Experiments show that the iterative generation of adversarial patches for loss function has a stable attack effect. Although humans can see the existence of a patch, it does not affect human cognition of the semantic information of the object image. Some algorithms have tried to optimize the appearance of the patch so that, when the patch is placed on the image, it will not feel like a "violation" due to the semantic correlation of the scene [25]. At the same time, it is noted that the adversarial perturbation and patch generated based on the white box setting are basically disastrous for the AI model. Without considering the scope, location, concealment and generation efficiency, the success rate of the attack has been close to 100%. Therefore, enterprises, such as automatic driving, face recognition and certification, should strictly control trade secrets, such as model structure and parameters, to prevent disclosure.

### 3.3. Instance Segmentation

Semantic segmentation is one of the key problems in the field of computer vision, which is very complex, from technical solutions to application scenarios. In general, semantic segmentation belongs to the task of high-level semantic understanding, which provides basic support for computer understanding of application scenarios. Scene understanding is a core task of computer vision. At present, many studies focus on extracting key semantics and temporal semantics in scenes because more and more top-level applications and programs need to obtain the basic knowledge of reasoning from the scene, such as automatic driving, virtual reality, security and other fields. With the wide application of deep learning, the depth and breadth of scene understanding have been further expanded.

The rise of deep learning has greatly improved the accuracy of semantic segmentation algorithms. In 2012, Ciresan used CNN to complete the task of semantic segmentation [26]. Sliding window is the main method; a small image block (patch) with each pixel as the center point is used as the basic unit. Ciresan input the small patch into CNN to predict the semantic label of the pixel. In the field of semantic segmentation, several algorithms based on region selection are gradually extended from the previous work on object detection to the field of semantic segmentation.

Based on the semantic segmentation model of an encoder-decoder structure, FCN completely discards the full connection layer in the image classification task and only uses the convolutional layer from beginning to end.

DeepLab v1, proposed in ICLR in 2015, is a method combining DCNNs and Dense-CRFs. The boundary of semantic segmentation is clear. The innovation of DeepLab v3 [1] is the improvement of the ASPP module [28]. The new ASPP module can aggregate the global context information, while the previous technology can only aggregate the local context.

Several methods have also attracted attention. For example, PSPNet uses pyramid-based modules or global pooling to regularly aggregate regional or global context information [28]. However, they capture the same kind of context but ignore different kinds of context. Additionally, attention-based methods, such as channel attention and spatial attention, selectively aggregate contextual information among different categories.

### 3.4. Object Detectors

The data includes N classifications, and the classification vector is represented by $h_c$. The function is used to describe the effect of the attack launched by attackers.

As a traditional task of computer vision, the object detection algorithm is mainly divided into two technical schemes. In 2016, Redmon J. et al. [3] proposed the YOLO algorithm. Many tasks, such as feature extraction, classification and recognition of objects in the image using YOLO, can realize the automation of image feature extraction and classification and recognition.

YOLO network structure is based on the GoogleNet model, which abandons the traditional method of manually labeling image features in the process of image recognition. In addition, RCNN has greatly improved the accuracy of object detection [29]. Since 2014, SPPNet* [30], namely spatial pyramid pool network, has enabled CNN to be expressed as a fixed length, avoid repeated calculation of convolution features and greatly improved the efficiency. Fast RCNN [31] has greatly improved speed and efficiency on the basis of Ref. [32]. Later, RFCN and other technologies significantly improved it.

The above technologies mainly adopt two types of networks: one-stage network and two-stage network. The anchor frame generated by a one-stage network is just a logical structure. The anchor box generated by a two-stage network can be mapped to the area of the feature map. The TS network re-inputs the region into the full connection layer for classification and regression. Each anchor mapped region needs such classification and regression, so it will be time-consuming. On the other hand, although there are not many anchors in the two-stage network, there are not many background anchors, that is, those unfavorable to network learning. Therefore, its accuracy is higher than that of one-stage.

In the fields of unmanned driving and smart city, it is necessary to quickly detect surrounding objects, so speed must be the primary consideration. Correspondingly, when attacking this kind of detection model, the accuracy and efficiency of the attack methods must be sufficient.

## 4. Our Methodology

### 4.1. Roadmap of the Proposed Approach

The adversarial attack is a research field of wide concern. Thus far, the existing methods can be divided into two categories. The first method is based on the adversary objective function set by attackers and uses the gradient to optimize the feature of clean data.

First, the attacker needs to obtain the parameters of the neural network model to be attacked, or obtain a substitute that approximates the target model. The saddle point function as the adversarial objective function should be set. The goal is to maximize the error output of the model and minimize the change of the feature of the optimized examples. Finally, through the output model function containing a neural network, the gradient method is used to search for the optimal solution for the object function.

Based on this theory, Ref. [1] proposed fast gradient method algorithm, which applies the first-order Taylor expansion to approximate the adversarial objective function, and it uses the one-step gradient descent method to find the adversarial examples. If the sign function is added to the gradient, multiplied by the step size, and updates the gradient, it is fast gradient sign method (FGSM). The two algorithms are fast, but the adversarial examples found are the first-order approximate optimal solution, not the optimal solution. Generally speaking, the example generation Equation (1) is as follows:

$$\eta = \varepsilon \cdot sign(\nabla_{x_{ori}} J(\theta, x_{ori}, y)) \tag{1}$$

Then, $x_{adv}$ is obtained by adding noise $\eta$ to the clean image $x_{ori}$.

$$x_{adv} = x_{ori} + \eta \tag{2}$$

Original images $x_{ori}$ adding perturbation $\eta$ need to be clipped for some out-of-range pixels. Although FGSM is a white box attack method, the adversarial example produced by FGSM also has a certain effect on the black box model, and it generally acts as a baseline.

The basic iterative method (BIM) algorithm performs several iterations on the basis of FGM and limits the part of the updated feature beyond the value to the domain. BIM is slower than FGSM/FGM, but it can find a better solution because it is no longer a one-step first-order approximation.

Szegedy et al. used L-BFGS to generate adversarial examples [2], which is a classic algorithm for box-constrained problems. C&W method [33] can generate strong adversarial examples based on this theory. Because the confidence of the generated examples can be adjusted, C&W can also attack the examples under both the condition of black box and white box.

The second method determines the input features that have the greatest impact on the output based on the Jacobian matrix between the model input and output and then changes these features to generate adversarial examples, such as Jacobian salience map algorithm (JSMA) [9]. Based on the optimization algorithm above, the adversarial example generation algorithm proposed in Refs. [32–34] is stable and has good transferability to different deep models. Refs. [4,33] proposed to select optimization variables and optimization methods for different data types. At present, the most effective attack method is still based on the first-order method.

Our algorithm is further optimized based on PGD. The random disturbance of the clean data in its neighborhood is used as the initial input of the algorithm, and the adversarial example is generated after multiple iterations. This algorithm initializes randomly before the attack, that is, adding random noise to the original image. The adversarial examples learned in each iteration are projected into the $\varepsilon - L_\infty$ neighborhood of clean data, so the adversarial perturbation is less than $\varepsilon$. At the same time, in order to meet the needs of different application scenarios, the strength of the algorithm in this paper is stronger than I-FGSM, and fewer pixels need to be modified.

### 4.2. Framework of the Proposed Approach

The algorithm proposed in this paper can generate adversarial examples that can deceive state-of-the-art object detectors. Its characteristic is that it can attack a specified object in the image so that the detector cannot recognize the specified object. The adversarial examples generated in this paper include two forms. The first one is perturbation, which is imperceptive to humans and is only with a small number of pixel modifications. When the adversarial perturbation is superimposed on the specified object in the image, humans cannot distinguish the adversarial example from the original image, but it can effectively mislead the detector. The second form is called adversarial patch. Adversarial patches do not need to consider whether they are visible or not. Patches generated in restricted areas can effectively hide specified objects and have a high attack success rate.

Figure 3 is the overall flow chart of this method. First, the image to be attacked is classified at the pixel level through FCN, and the category of each pixel is recovered from the abstract features. The algorithm establishes a semantic category and instance mask and then extracts the specified object from the image. Then, we select the hidden object to attack through the selector and generate its corresponding mask. These masks will be used as restricted areas for training adversarial examples. Further, by iteratively updating the objectness scores of the loss function within the restricted area, the adversarial examples are continuously optimized and generated.

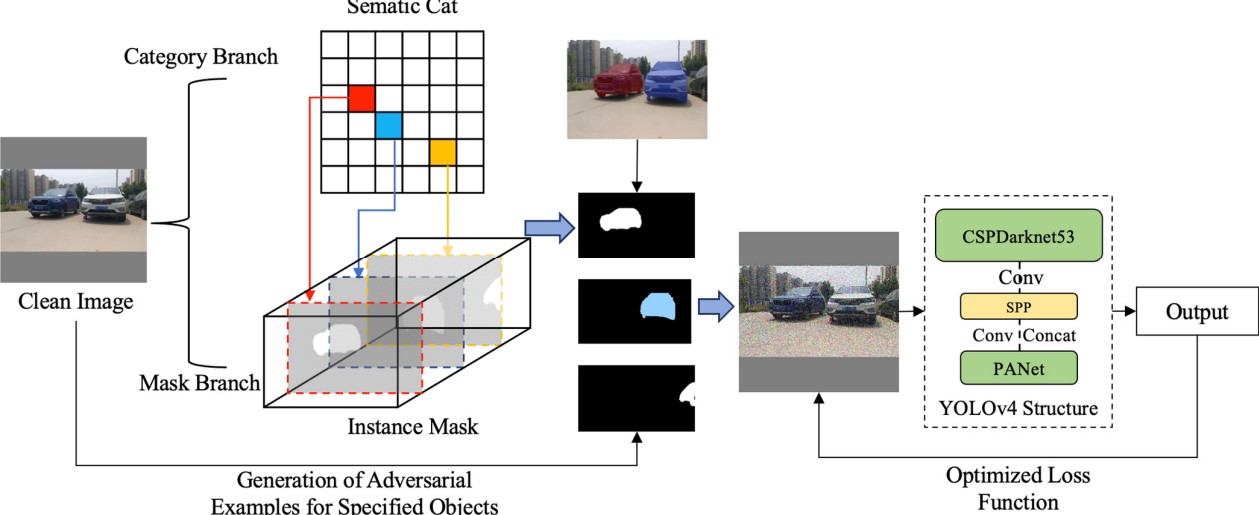

**Figure 3.** The framework of the proposed method. As shown in the figure, the algorithm consists of two main parts. The first part is the generation of adversarial examples for specified objects. In this part, the image is cut based on the instance segmentation algorithm. The mask defines the generation region for adversarial perturbations. The second part is a combination of a series of optimized loss functions. The full image perturbation is optimized within the restricted region, and new adversarial examples are generated.

### 4.3. Overview of the Optimization Function

With the continuous development of deep learning technology, the robustness and accuracy of object detectors based on deep model are also improving. The goal of this paper is to generate adversarial examples in a specified range that can effectively deceive the object detector. As mentioned earlier, there have been successful cases of mixing adversarial examples with clean images to generate toxic data [4] for training new deep models. The content of the image is hidden after adding perturbation to the whole image. This paper finds another way to improve the application field of adversarial attacks by generating adversarial examples in specified areas.

#### 4.3.1. Overview of the Optimization Function

Softmax loss is a combination of Softmax and crossentropy loss. This method is widely used in image classification and image segmentation tasks. In the mainstream deep learning algorithms, researchers combine the two contents into one layer for calculation, rather than calculating them as two independent modules, which can make the numerical calculation more stable. After adding an adversarial patch, the goal of this paper is to calculate the total loss and find the minimum value by adding noise to the prediction as close as possible to the object. The final total loss function is optimized by multiple loss functions, where $L_1$ formula is as follows:

$$L_1 = arg_r min \sum_{x \in X} softmaxloss(f(x+r), l) \tag{3}$$

Equation (3) shows that, for the target object, the noise to the input image $x$ needs to be found through the optimization function. Assuming that the perturbation is $r$, the probability of classification $l$ after increasing the perturbation is maximum. The distance between $f(x+r)$ and $h_t$ is calculated (1), which is calculated by function $L_1$ in this paper.

$L_2(p)$ formula is used to calculate that the generated patch fits the placed region as completely as possible. Considering that the pixels of future pictures may not be high enough, or another noise is introduced when output is required, we improve the attack effect through optimization and prevent the impact caused by noise. This situation is

discussed in Ref. [34]. For optimal results, the lower the score, the closer the pixels of the generated patch are to the pixels placed in the image. This function improves the robustness of the attack and can adapt to more images and environments.

$$L_2(p) = \sum_{i,j} \left( \left( p_{i,j} - p_{i+1,j} \right)^2 + p_{i,j} + p_{i,j+1}{}^2 \right)^{\frac{1}{2}} \tag{4}$$

$L_3$ is used to calculate the score of the detection frame of the detector during detection. The object detector generates a bounding box in the image to obtain a prediction score. The score of this part is generally composed of two parts; one is the objectness score and the other is the class score. This function makes the object detector unable to detect a specified object by reducing the objectness score ($L_3$ score) of the detector.

To sum up, the total loss function used to generate an adversarial patch in this part is as follows:

$$L_{total} = \alpha L_1 + \beta L_2(p) + \gamma L_3 \tag{5}$$

This algorithm comprehensively considers three loss functions and adjusts the total loss function through three parameters. The parameters $\alpha$, $\beta$ and $\gamma$ are given by expert opinions, and we use Adam algorithm as in Ref. [34] used for optimization. In order to reduce the influence of network structure on model parameters, all weight variables are fixed, and the value of the adversarial patch is adjusted to obtain the optimal patch.

Four forms of adversarial patches generated by different methods are given in Figure 4. Figure 4a shows the adversarial patch generated based on $L_{total}$ in this paper. Figure 4b is the adversarial patch generated in Ref. [23], which can attack the whole picture. Object detector is totally blind. Figure 4c is a printable patch that can attack all objects in the real world [21]; Figure 4d is a patch called Dpatch that can attack the detector with a very small size [26].

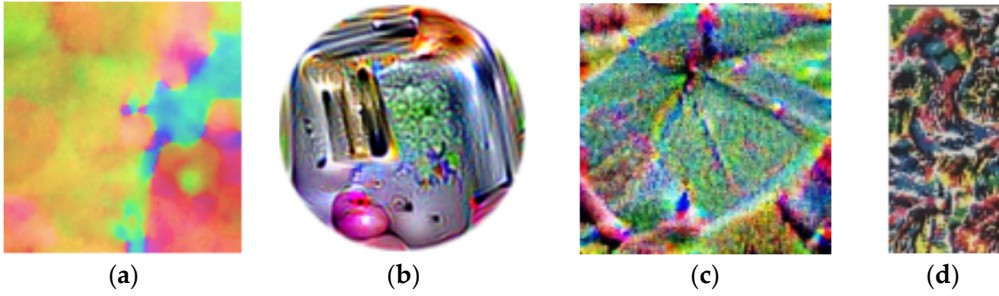

(a)     (b)     (c)     (d)

**Figure 4.** The morphology of the adversarial patches is generated by different algorithms. (**a**) Adversarial Patch from our paper. (**b**) Universal Adversarial Patch. (**c**) Printable Adversarial Patch. (**d**) Dpatch.

### 4.3.2. Restricted Adversarial Perturbation Generation

In this paper, the PGD algorithm is for reference, and the adversarial examples in the restricted area are generated after clipping based on the discussion of adversarial examples strategy in Ref. [6]. In a defense effect test, researchers set attack parameters through the white box, and the PGD algorithm is the only one that has stood the test. The algorithm randomly initializes the attack data in the image range and generates adversarial information through iteration. The algorithm uses the generated adversarial perturbation for adversarial training. The first step is to use noise pixels as initialization content.

In the process of multiple iterations, each small step will limit the perturbation to the specified range through clip operation. This operation is to prevent the range of each step from being too large, which improves the success rate of attacks. Therefore, this paper

adopts the Improved algorithm of PGD as the attack algorithm, and the mathematical description is as Equation (6):

$$x^{t+1} = \prod_{x+S} \left( x^t + \alpha \cdot sign(\nabla_x L(\theta, x, y)) \right) \tag{6}$$

In the Equation (6), $x \epsilon R^d$ is the input picture and is a group of adversarial perturbation. The perturbation used in this paper is *l∞ball* constructed around the input image *X*. *L* is the loss function, where $\theta$ is the model parameter and sign is the *Sign* function. We set the super parameter $\alpha = 0.01$. Further, in the process of each iteration, the pixels in the image are clipped, and the updated Equation (7) becomes:

$$x := clip_{[0,1]}(x + \alpha \cdot sign(\nabla_x L(\theta, x, y))). \tag{7}$$

After the clip function is added, the range and color of the generated perturbation can be effectively controlled, as shown in Figure 5 below. Figure 5a shows the adversarial examples generated without clip function; Figure 5b shows the adversarial examples with clip function.

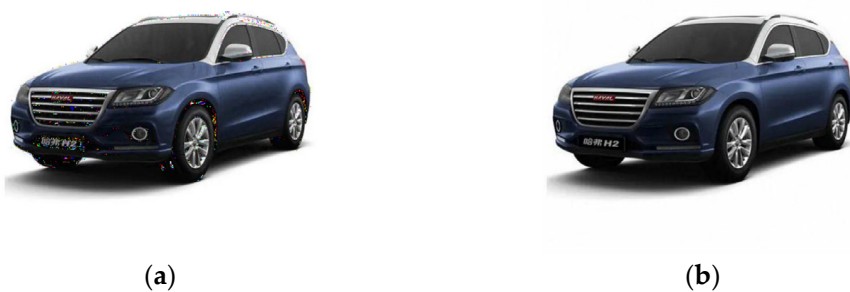

(**a**)                                               (**b**)

**Figure 5.** Comparison of the effects of unclip and clip function on adversarial perturbation generation. (**a**) Adversarial perturbation without clip function. (**b**) Adversarial perturbation with clip function. Careful identification of the two images will reveal that colorful noise can be clearly seen in the light-colored part of the car in (**a**). The perturbation information is also present in the dark part, but it is covered by the body color. By superimposing the adversarial perturbations processed by the function into the images, a more hidden adversarial example can be generated.

4.3.3. Restricted Adversarial Perturbation Generation

YOLOv4 is a single-stage object detector with excellent speed performance and considerable accuracy performance. On the basis of YOLOv3, under the condition that FPS does not decrease, the mAP of YOLOv4 reaches 44, which is significantly improved. YOLOv4 uses three feature layers for classification and regression prediction. It uses the confidence of the candidate box to distinguish the foreground and background in the image. If the confidence of the candidate box is lower than the threshold, the object will be recognized as the background and discarded in the following stage.

The output of YOLO detector is a probability, which judges whether an object is included in the image. At the same time, it generates a class score to display which categories are included in the bounding box. Based on the above principles, this paper designs a loss function of YOLOv4 attack by reducing the confidence of the candidate frame. The function is as follows:

$$loss_{yolo} = \sum_{i \in \{j|s_j > t\}} (\alpha \cdot obs_i + \beta \cdot cs_i) \tag{8}$$

where $obs_i$ is the score of each bounding box generated by the detector, which is used to indicate whether the box contains the object to be detected; $CS_i$ is the class score, which is used to judge which category the object contained in the box belongs to. Two parameters, $\alpha$ and $\beta$, are set according to expert experience; t is the attack threshold in the calculation

process, which is set to 0.2 empirically in this paper. During the attack, the algorithm continuously evaluates the bounding box score and class score of the specified object output by the detector and continuously optimizes to achieve local or global optimization.

In the training process, the goal of this paper is not only to generate deceptive adversarial examples but also to limit the generated adversarial examples to specified areas. In this paper, the object to be attacked is extracted based on semantic segmentation algorithm, and the attack location is guided by mask. The algorithm only restricts the adversarial examples in the mask area and continuously reduces the perturbation data outside the area. The generation Equation (9) used is as follows:

$$x^{t+1} = \sum_{x+S} \left( x^t \odot (1 - Area_m) + p_{adv} \odot Area_m \right) \tag{9}$$

where $x^t$ is the clean image to be attacked; $p_{adv}$ is the adversarial perturbation generated from the whole image. It means the generated adversarial example, in this paper, can be either adversarial perturbation or adversarial patch. $Area_m$ is the color scale matrix of the restricted region generated according to the semantic segmentation algorithm. The matrix is in the form of 0/1, with 0 for black and 1 for white.

The calculation process is shown in Figure 6. YOLOv4 uses the confidence of the object candidate box to distinguish between foreground and background. If the credibility is lower than a certain threshold, the candidate box will be recognized as the background and will be discarded afterward.

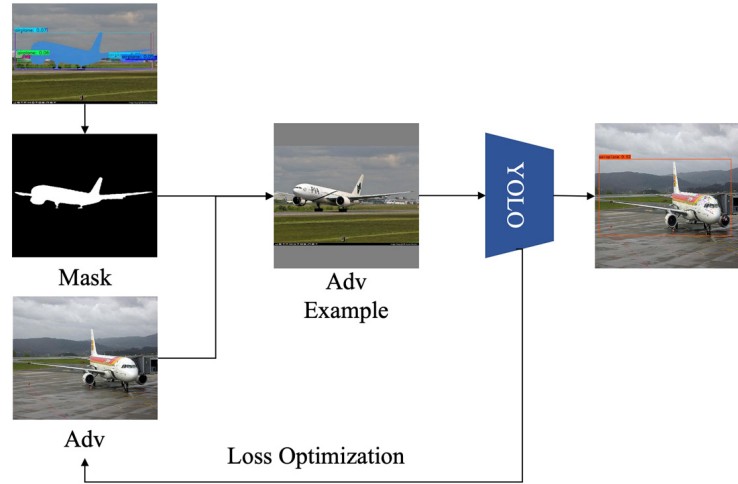

**Figure 6.** Restrained adversarial examples generation inside masks.

## 5. Experiments

### 5.1. Experiment Configuration

The method proposed in this paper can be widely used in the deep learning model attack of automatic driving object detection, medical diagnosis and field of animal and plant object recognition. In the process of deploying the attack model, the goal of this paper is to attack the mainstream detectors. Because our adversarial perturbation is generated within a restricted area, we call it restricted area adversarial examples generation algorithm (Re-AEG).

All models in this paper are trained and tested in the open dataset, in which the ratio of training set to test set is 8 to 2. Two types of public data and attack methods are used to test. Among them, ImageNet dataset is adopted in the multi-class object detection and attack method competition. It includes more than 14 million images and is suitable for the use of object localization, object detection, object detection from video, scene classification, scene parsing and other scenes.

The second dataset is coco2017, which is developed and produced by Microsoft [35]. This dataset has in-depth applications in the fields of object detection, object segmentation, object localization and captioning. The version used in this paper includes 80 categories, including 120 thousand training set images, 5 thousand verification images and 40 thousand test set images.

### 5.2. Baseline Attacks and Ablation Study

The innovation of this paper is to limit the generated adversarial examples to specific areas so that the adjusted traditional methods can attack specific objects in digital images. This paper does not test physical examples because the print color gamut and the position and size of the restricted area are too complicated to generate, which is not discussed as a part of this paper. Tests are operated on ImageNet [35] and coco2017 [34] datasets and compare the effects of different factors on the effectiveness of adversarial examples and patches, which include the position and shape of the defined area in the image. The threat model in this paper is not carried out in a complete white box test setting. The backbones of the networks in both offensive and defensive sides adopt Darknet-53 [28] as the feature extraction network architecture. Compared with the popular ResNet [30], although the classification performance is deficient, the object detection accuracy of CSPDarknet53 [28] is higher and more suitable for testing adversarial examples.

### 5.3. Adversarial Attack on Simple Scene

#### 5.3.1. Experiment Description

This section verifies the effectiveness of the basic functions of the algorithm by setting parameters and models. We test four models: VGG16/19 [18], Xception [19] and inception-v3 [21] based on the training method in Ref. [21] and train the perturbation through multiple ImageNet using white box setting. The basic idea is to generate adversarial examples by restricting the area of perturbation and to deceive the object detector in large quantities and quickly.

According to research [32], the white box attack can be realized by modifying only one pixel of the picture in CIFAR-10 dataset. However, compared with the attack against specific objects in this paper, simply reducing the number of pixels that need to be modified is not easy to popularize in practical situations. Studies [34] prove that, although the success rate of the adversarial examples generation algorithm for the whole image is very high, the adversarial process does not include the process of object detection and recognition in the image. First, the algorithm identifies the subject-object in the image, identifies the object through the instance segmentation algorithm and establishes the corresponding mask matrix. Finally, it carries out selective attacks in combination with application scenarios.

#### 5.3.2. Experiment Results

The first set of experiments shows the attack effect in a simple scene. The experiment is mainly divided into three groups: images containing only one object; images containing two objects; images containing multiple objects.

As shown in Figure 7, the clean image contains only one aircraft in a cloudy environment. The algorithm extracts the aircraft from the picture and generates adversarial examples in the constraints of the mask matrix. YOLOv4 cannot detect the aircraft, as predicted. The second set of pictures contains two aircraft. In order not to attract the enemy's attention in a combat environment, we only attack one of the fighters. The adversarial examples are generated by referring to the position of the mask matrix, and the detector can only detect the second aircraft but ignore the first aircraft. The third group of pictures tried to test more complex life scenes. Based on the instance segmentation algorithm, multiple objects in the image were separated, and then the attack mask was generated for the attacking portrait. Finally, the girl who only showed half her body was successfully hidden.

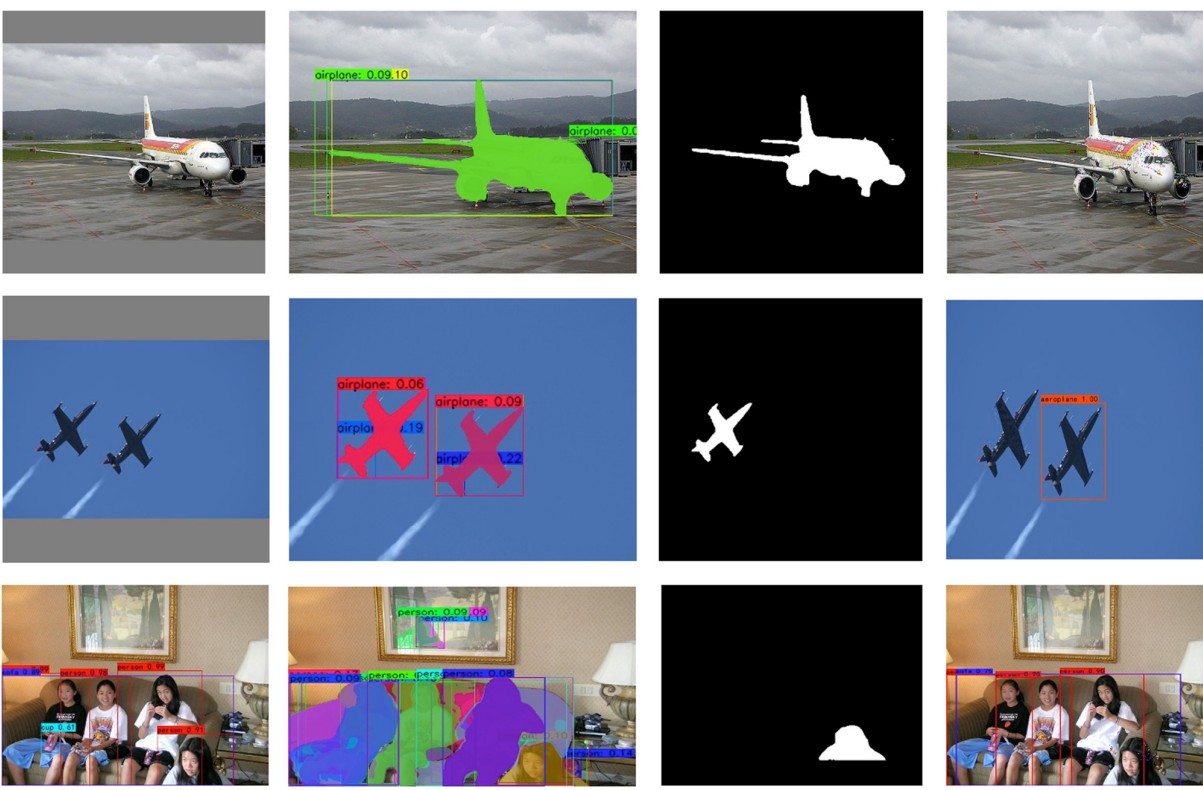

**Figure 7.** Comparison of adversarial examples to images with a different number of objects.

### 5.4. Adversarial Examples in Complex Environments

In order to verify the effectiveness of the algorithm, traditional methods often generate adversarial examples on clean images with a single object. However, the conditions in the real scene are much more complex, such as complex background, more objects, overlapping and covering between objects, etc. The example in this section is a picture of seven cyclists taking a group photo on the road, attacking one of the cyclists. This algorithm implements two types of adversarial examples: the first is to generate an invisible adversarial perturbance within the restricted region; the second form is to generate adversarial patches accordingly. In the experiment, this paper tests the attack effectiveness of the algorithm by restricting the scale and shape of the adversarial area. As shown in Figure 8, the attack effectiveness of generating adversarial examples through the areas of different parts of the body of the first rider on the right is tested, respectively.

This paper detects the objects in the image through YOLOv4 and circles seven people and six bicycles in the image through the bounding box, which shows that the detection model used to test the algorithm can work efficiently. In the second step, the candidate objects of all attacks are drawn through the instance segmentation algorithm. Figure 8c shows the contour of the seven people segmented. The system automatically locates the attacked objects according to the requirements of the application scenario. In order to analyze the impact of the scale and location of the restricted area on the attacked objects, three different mask matrices are divided in Figure 8c. From top to bottom, there are complete human masks, human upper body masks and human lower body masks. This paper generates adversarial examples based on these three different mask matrices. In order to clearly show the attack area, we outline the area where the attacked part is located with a white dotted line in Figure 8e.

Figure 8f shows the effect of the algorithm proposed in this paper to generate adversarial perturbance in the restricted area, and Figure 8g shows the effect of the adversarial patch.

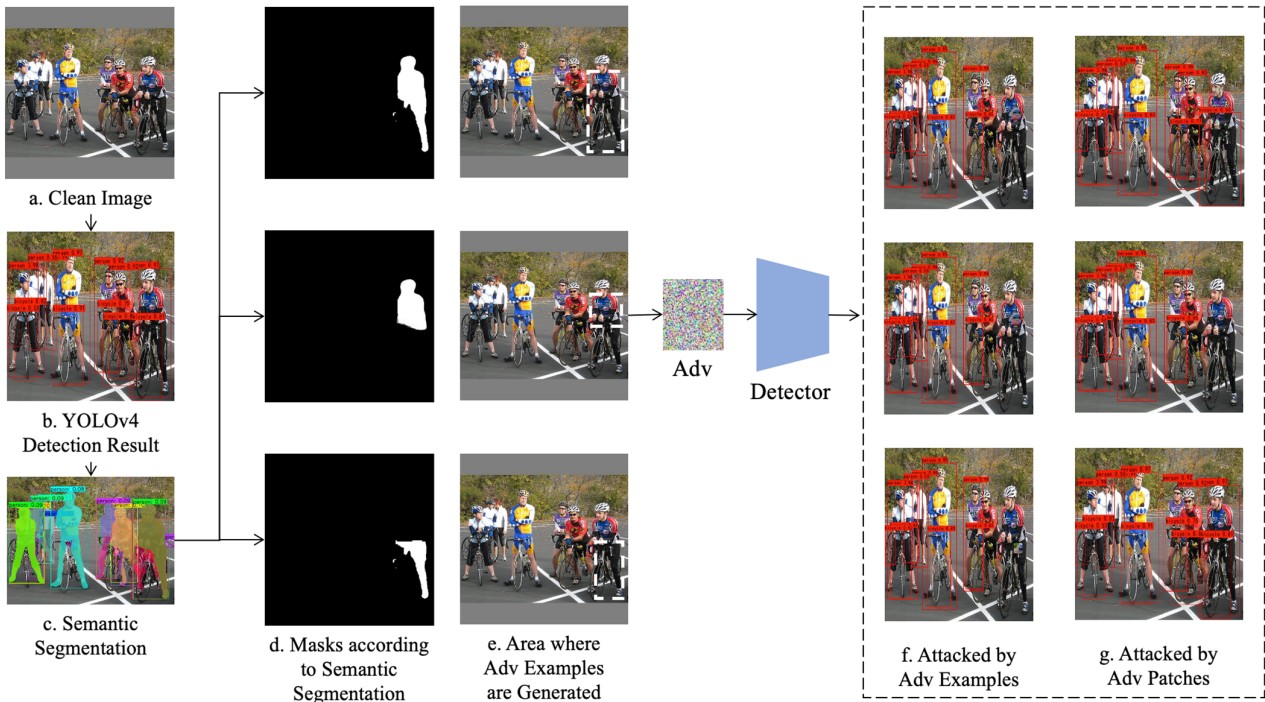

**Figure 8.** Ablation study of adversarial examples in different locations and sizes.

This paper still detects the other objects after the attack through YOLOv4, and the generated adversarial examples work efficiently. However, since the target object is not for a complete image of the whole area, we can see the adversarial patch not only hides the cyclist but also makes his bicycle undetectable. It means the attack effect overflowed. In the last image, because the attack area is too small with a very complicated background, after forty thousand iterations, our algorithm does not converge, and the detector is not being deceived.

*5.5. Analysis of Attack Accuracy*

In this section, we quantitatively test the attack strength, convergence rate and optimization of the algorithm. The perturbation capability $\varepsilon$ in the algorithm indicates how much the algorithm can modify the pixels in the image so as to induce the detector to make mistakes. The number of iterations $\mathcal{T}$ determines the optimization time of the algorithm in a real attack scene. While attacking large-scale images, it is often necessary to strike a balance between attack success rate and attack speed. In the process of algorithm optimization, the convergence rate is controlled by adjusting the optimization step $\alpha$.

We use four training models for testing in this paper: Inception_V3(IncV3) [21], VG-GNet16 [19], ResNet-50 (Res50) [30] and Inception Resnet_v2 152 (IncRe152) [30]. We selected 2000 images from the ImageNet dataset for training and 1000 images as the baseline of the control group. The attack accuracy is tested in four classical adversarial example generation algorithms. The matrix is calculated as follows:

$$Acc(C, \mathcal{A}_{\varepsilon,p}) = \frac{1}{N} \Sigma_{i=1}^{N} \mathbb{1}\big(C(\mathcal{A}_{\varepsilon,p}(x_i)) = y_i\big) \tag{10}$$

where $C$ is the target classifier, $\mathcal{A}_{\varepsilon,p}$ is the adversarial example generation module, $\varepsilon$ is the attack capacity and $p$ is $\ell_p$ norm. The success rate calculated in the Equation (10) is equal to the number of successful adversarial examples divided by the total number, which indicates the extent to which the algorithm can generate effective attacks on the target model. The more adversarial examples that can successfully deceive the misclassification of the deep model, the more successful the attack method is. We compare the algorithm

proposed in this paper with FGSM, BIM and ILLC algorithms. Based on the fixed attack intensity coefficient and norm, we test the attack success rate for different models, and the results are shown in Table 1.

**Table 1.** Detection accuracy of different models.

| Models | Clean Images | Re-AEG | FGSM | BIM | ILLC |
|---|---|---|---|---|---|
| VGG16 | 94.83% | 1.37% | 2.09% | 5.44% | 16.94% |
| IncV3 | 97.22% | 1.64% | 6.16% | 5.45% | 4.13% |
| Res50 | 95.31% | 2.88% | 16.74% | 16.33% | 13.08% |
| IncRes152 | 90.67% | 1.28% | 7.87% | 4.48% | 2.03% |

In addition, this paper also tests the influence of perturbance capacity (epsilon) hyperparameters on the performance of the algorithm, as shown in Figure 9. Four attack methods of four models under the ImageNet dataset are tested. The *x*-axis is the setting of perturbance capacity (epsilon), which represents the attack intensity, and the *y*-axis is the accuracy. With the increase in the parameter epsilon, the attack success rate against the examples increases continuously. In order to reduce the impact of environmental differences and other factors, the blue dotted line in the figure indicates the success rate of model detection of clean images. The steeper the curve decreases, the more effective the attack is. Among the four attack methods tested, the algorithm Re-AEG proposed in this paper can achieve a better attack success rate with smaller perturbation amplitude in a short time.

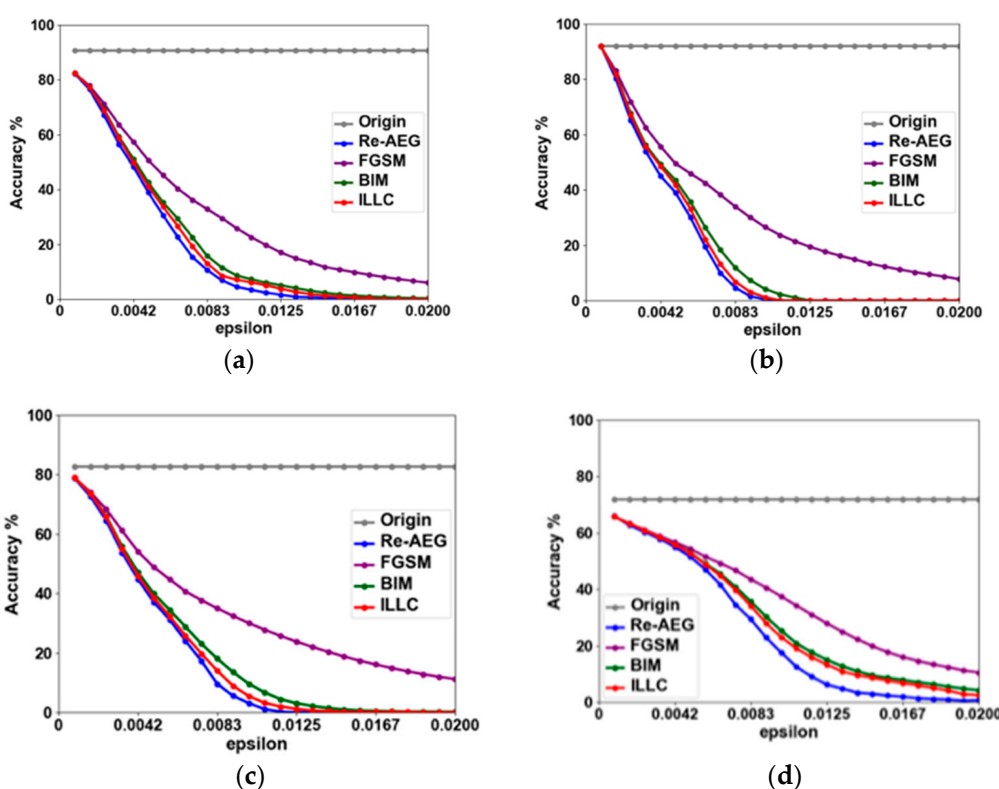

**Figure 9.** Comparison of different models. (**a**) Inception; (**b**) VGGNet16; (**c**) ResNet-50; (**d**) IncRes152. In this paper, the accuracy of the proposed algorithm is tested against three other classical adversarial algorithms for each of the four commonly used deep models. It is proved that our algorithm is able to obtain a higher success rate with lower attack strength.

This paper tests the balance between the number of iterations and the loss function of Re-AEG algorithm. As shown in Figure 10, the loss function jitters in the later stage of the training process. The introduction of momentum will reduce such fluctuation. It is

found that, due to the different areas of adversarial examples and optimization functions, the attack success rate of adversarial perturbation is significantly higher than that of adversarial patch.

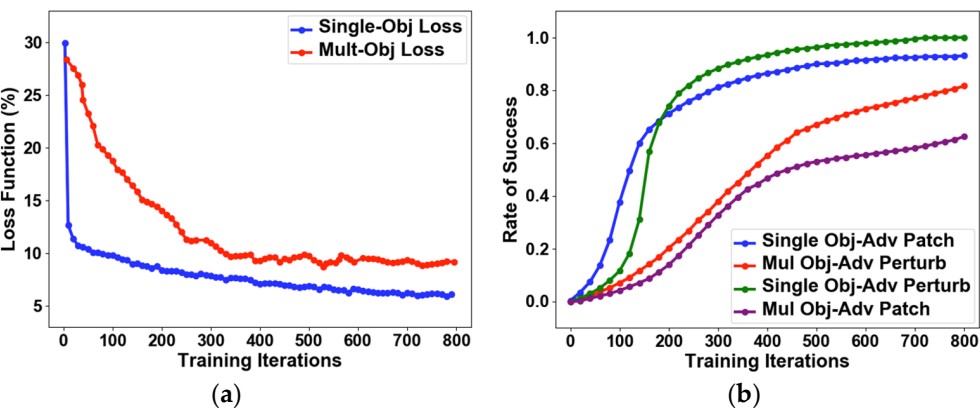

**(a)**　　　　　　　　　　　　　　　　**(b)**

**Figure 10.** Performance based on training iterations. (**a**) Loss comparison curve of a single object and multi-object adversarial examples. (**b**) Comparison curve of the success rate of a single object and multi-object adversarial examples.

Figure 10a also shows that, in the process of attacking different numbers of objects, our algorithm generates adversarial examples for each object and sums them. Therefore, the algorithm keeps calculating the minimum value according to the lower limit, resulting in fluctuation of the curve. Figure 10b tests the attack success rate of the algorithm against a single object and multiple objects by generating perturbation and patches.

### 5.6. Influence of Parameters on Algorithm

This paper tests the value of the main parameters in the process of example generation. We set parameter $\varepsilon$ in Re-AEG to limit the $L_\infty$ distance between images before and after attack. In this paper, the proposed attack method is tested on different object detectors, and the success rate is calculated. Table 2 shows the accuracy of the algorithm proposed in this paper when attacking different object detectors.

**Table 2.** Detection accuracy of different object detectors.

| Models | ImageNet-Single | ImageNet-Multiple | COCO-Single | COCO-Multiple |
|---|---|---|---|---|
| YOLOv4 | 94.23% | 92.63% | 95.75% | 95.52% |
| Faster-RCNN VGG16 | 94.29% | 92.79% | 96.01% | 95.63% |
| R-FCN ResNet101 | 93.66% | 93.28% | 95.83% | 95.86% |
| YOLOv4 | 94.23% | 92.63% | 95.75% | 95.52% |

This section may be divided by subheadings. It should provide a concise and precise description of the experimental results, their interpretation, as well as the experimental conclusions that can be drawn.

Through experiments, this paper analyzes the influence of detection parameters on the attack effect. By assigning different parameters, the changing trend of the attack strength of the algorithm is studied. In this paper, the parameters $\varepsilon$ are 8/1, 4/1 and 2/1, respectively, and $log_2\varepsilon$ is used to show the impact of parameter $\varepsilon$ on the accuracy of the attack, as shown in the following Table 3.

**Table 3.** Accuracy of the deep model by different attack parameter $\varepsilon$.

| $log_2\varepsilon$ | Accuracy | $log_2\varepsilon$ | Accuracy |
|---|---|---|---|
| −3 | 98.95% | 3 | 52.69% |
| −2 | 98.95% | 4 | 56.49% |
| −1 | 99.18% | 5 | 62.49% |
| 0 | 98.58% | 6 | 63.46% |
| 1 | 94.86% | 7 | 62.94% |
| 2 | 76.24% | 8 | 62.79% |

The experiments show that, with an increase in attack strength, the attack accuracy is divided into two stages. In the first stage, the accuracy of the detector continues to decline, and the success rate of the attack increases. However, in the second stage, the attack effect jitters. With the increase in attack strength, the step size of a single attack is too large, but the model capacity remains unchanged, resulting in a decline in attack success rate. This is consistent with the conclusion in Ref. [33].

**6. Conclusions**

Adversarial examples challenge application of the AI deep model. Its theoretical basis is becoming more and more mature and complete, but, in the process of practical application, due to its various generation methods, it is almost impossible for security personnel to prevent. The object detection system based on deep learning is powerful, requires a small amount of human participation and greatly reduces the costs of time and money. However, in medical-, financial-, military- and civil-safety-related fields, such as automatic driving, it is unacceptable to fail to ensure the safety of the correct operation of the system. The high rate of return has also spawned more and more high-tech crimes. Therefore, it is of great practical significance to carry out attack and defense training of the deep learning model.

First, this paper analyzed the principle of generating adversarial examples in the actual scene. Artificial intelligence is widely used in security-related fields, such as finance, banking and transportation. Hackers and counterfeiters are more likely to attack the deep model through data fabrication. In view of this kind of security hidden danger, this paper proposed an adversarial examples generation algorithm (Re-AEG) to attack specific objects in digital images. The core idea of the algorithm is to accurately attack one or more specific objects in the image, which affects the accuracy of detection. In the implementation process, this paper first realized the adversarial perturbation algorithm generated in the restricted area, which can quickly converge with a high success rate and mislead YOLOv4. On the other hand, this paper tested the generation algorithm of an adversarial patch in a restricted area. The experiments showed that the success rate of the latter depends on the size of the target object and the ratio of generated patch to target object. The attack effect of the generated patch is not as obvious as that of adversarial perturbation.

The experiment in this paper was carried out in the digital domain. One of the reasons is that the adversarial examples are limited to a certain local area. This area accounts for a small proportion relative to the whole image, and the number of pixels modified by the generated perturbation is smaller. Therefore, if a physical attack is implemented, the error (noise pixels) generated by the scanner or printer may be greater than the generated adversarial examples.

Different from the traditional malicious program attacks, an attack on the deep model occurs in the stage of deploying models, which is more difficult to defend against and find. The research of this paper integrated two basic forms of adversarial examples to attack digital images flexibly. This study is of great significance to improve the robustness of the deep model.

**Author Contributions:** Investigation, Y.H.; Methodology, D.L.; Software, S.L.; Writing—original draft, D.C.; Writing—review & editing, Y.H. All authors have read and agreed to the published version of the manuscript.

**Funding:** This research was funded by the project of SASTIND, PRC, Grant No. JCKY2021206B102.

**Institutional Review Board Statement:** Not applicable.

**Informed Consent Statement:** Not applicable.

**Data Availability Statement:** The raw data coco2017 required to reproduce the above findings are available to download from https://cocodataset.org (accessed on 2017). And the ImageNet dataset can be download from https://image-net.org.

**Conflicts of Interest:** The authors declare that they have no conflicts of interest to this work.

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
