# Peer review of "An Adversarial Attack Method against Specified Objects Based on Instance Segmentation"

_information, doi:10.3390/info13100465_

Round 1
Reviewer 1 Report
This work proposes an edge-restricted adversarial example generation algorithm based on the instance segmentation. This algorithm can attack one or more specific objects in the image so that ther detector cannot detect the objects. Experimental results on ImageNet and COCO2017 datasets demonstrate the proposed method can successfully attack single or multiple objects in multi-object and complex senes. Overall, the technical contribution of this work is moderate. The detailed comments are as follows:
1. For the title “Adversarial Attack Methods Against Specified Objects based on Semantic Segmentation”, we would like to suggest that the author revise the title to “Adversarial Attack Methods Against Specified Objects based on Instance Segmentation”, because the masks of instance segmentation are mainly used in this work.
2. For the Figure in Page 3, Figure 1 should be Figure 2. In Figure 2, 3, and 5, the caption should be revised. Besides, the quality of all figures should be enhanced.
3. In P7308, the author states “we select the hidden object to attack through the selector …”. How to select the hidden object and what is the structure of the selector?
4. What is the instance segmentation method used in this work?
5. For the testing, 2,000 images are selected from the ImageNet dataset for training, and 1,000 images are selected as the baseline. Are there results from a common public testing set.
6. There exist some spelling issues,
a) In P6L264, the index of the equation is missing.
b) In P9L375, “formula 7” should be “formula 6”.
c) In P11L465, P11L468, “research 22” and “Studies 25” should be correctly cited.
Author Response
Dear Reviewers,
Thanks very much for taking your time to review this manuscript. I really appreciate all your comments and suggestions.We have tried our best to improve and made some changes in the manuscript.Revision notes, point-to-point, are given as follows:
Point 1:For the title “Adversarial Attack Methods Against Specified Objects based on Semantic Segmentation”, we would like to suggest that the author revise the title to “Adversarial Attack Methods Against Specified Objects based on Instance Segmentation”, because the masks of instance segmentation are mainly used in this work.
Response 1:
The Comments posed by the reviewer is very accurate. In this paper, the algorithm attacks multiple objects and must separate the attacked objects by instance segmentation algorithm. And one or more objects are hidden.
Point 2:For the Figure in Page 3, Figure 1 should be Figure 2. In Figure 2, 3, and 5, the caption should be revised. Besides, the quality of all figures should be enhanced.
Response 2:
(1) According to the opinions of the reviewer, we switch Fig. 1 and Fig. 2 . And accordingly, we unpdated the meaning and captions of the two figures.
The logic of the revised text is as follows: first, the basic flow of the algorithm is introduced. Seconde we compare with the traditional adversarial example generation algorithm with the traditional methods by attacking the same picutures. Third, the intuitive effect of the algorithm is introduced.
(2) On the basis of updated the pictures, authors improved the titles of figures 2, 3 and 5. Through the caption of the figure, the meaning and process of the picture are introduced. Please refer to the corresponding position of the original manuscript for the specific modification.
(3) Image modification.
In the original manuscript, due to the limited length of the paper, the author reduced the size of the picture when inserting the picture, resulting in the text in the picture is also smaller, reducing the readability of the picture. Thanks for reviewer's reminder that the author has optimized the picture and restored its original size. For details, please refer to the corresponding position of the original picture.
Point 3:In P7308, the author states “we select the hidden object to attack through the selector …”. How to select the hidden object and what is the structure of the selector?
Response 3:
The selector used in this paper has two functions: the first one is to select the object with attack from multiple instance segmentation algorithms. This process can be completed by human operator; The second can be used to delineate the area in the picture that is used for attack. The instance segmentation algorithm can separate the attack object from the delineated area.
In an attack scenario, the operator can select an area or target to attack through the Selector application. This operation, there is not much worth describing academically, so it is not detailed in the paper. However, in a real attack scenario, this operation can be of great practical value as part of the adversarila examples attack, allowing the operator to quickly specify the area to be attacked (the area containing the objects). A rendering of the selector implemented in this article is shown below:
The airplane (pictured above) -001-007.jpg is a manually drawn mask for a single object; Running the instance segmentation algorithm in this area can separate the attack objects more accurately. In other images, there are 2-8 areas with different numbers, sizes, and locations. These indicators are determined by the operator in practice, and the selected area will become the area for generating adversarial samples in the future. Images that are entirely white are compatible with the traditional attack method, which is adversarial examples generated from the whole image.
In summary, the goal of this paper is to generate effective adversarial perturbations based on arbitrarily formulated regions (shapes of people, objects) without strict constraints on the structure or principle of selectors. Therefore, the code details of selector are not included in the paper.
Point 4: What is the instance segmentation method used in this work?
Response 4:
In this paper, an edge-restricted adversarial example generation algorithm is proposed, and the function of instance segmentation algorithm is to accurately locate the object to be attacked in the image to be attacked. Based on the explanation of selector in point 3 above, the accuracy of the Instance Seg algorithm is not strictly required in this paper, but in practical applications, the faster the segmentation, the better.
In this paper, we train multiple instance segmentation models on miniImageNet, a subset of the dataset ImageNet, and implement the segmented objectives. These instance segmentation algorithms include: SOLOv2, Mask R-CNN, PolarMask, and BlendMask. The final choice was SOLOv2. For the following reasons:
first, the SOLOv2 framework is more lightweight and simpler to configure among several algorithms;
Second, SOLOv2 is the one with the best balance between speed and AP. The AP can be improved by 6% at comparable speeds.
Point 5:For the testing, 2,000 images are selected from the ImageNet dataset for training, and 1,000 images are selected as the baseline. Are there results from a common public testing set.
Response 5:
We trained a YOLO model for our experiments on ImageNet, but we didn't list the detection results for this model. The reasons are as follows:
First, the YOLO model is not trained to improve the detection efficiency of the model, but to use the feature extraction and recognition ability of the model as a baseline to compare with the post-attack detection results.
Second, The effectiveness of the attack method can be evaluated as long as the accuracy decrease of the image after being attacked.
Third, the accuracy of YOLO does not necessarily have a strong reference significance due to the differences in the specific code and server Settings, so the test detection results listed in the paper cannot prove the effectiveness of the attack algorithm.
Therefore, our aim is to obtain the comparison between pre-attack and post-attack, rather than to obtain the optimal performance of the detector.
Point 6: There exist some spelling issues,
- In P6L264, the index of the equation is missing.
- In P9L375, “formula 7” should be “formula 6”.
- In P11L465, P11L468, “research 22” and “Studies 25” should be correctly cited.
Response 6:
We are seriously sorry for the problems in format and grammar. The authors made the following changes:
- A) Supplement the index of the equation (1)
- D) "Formula 7" has been modified to "Formula 6".
- B) The questions of "Research 22" and "Studies 25" have been modified.
Other formatting issues were also modified.
Thank you for your time and comments.

Reviewer 2 Report
I recommend that the Motivations section becomes part of the introduction by transforming it into an objectives and research questions. It is necessary to expand the number of cited sources in the introduction by expanding its size.
Author Response
Main Point:
I recommend that the Motivations section becomes part of the introduction by transforming it into an objectives and research questions. It is necessary to expand the number of cited sources in the introduction by expanding its size.
Dear Reviewers,
Thanks very much for taking your time to review this manuscript. I really appreciate all your comments and suggestions.We have tried our best to improve and made some changes in the manuscript.Revision notes, point-to-point, are given as follows:
Response 1: We merged the Motivations into the Introduction section, and reorganized and organized the logical relationship between the cited documents.
We study the revised Introduction section. Starting from the application scenario of the deep model, the background and principle of the generation of adversarial examples are analyzed, and the objectives of the paper are analyzed.
Then, combining with the existing algorithms and research progress, this paper analyzes the current research hotspots and research divisions, and proves the rationality of the proposed algorithm and research content.
Response 2:
Combined with the reviewer’s comments, we have added some literature related to research, application and existing problems in the Introduction section.
The authors also discuss some key aspects of the proposed algorithm in detail. For example, adversarial examples are an important reason why deep learning applications cannot be promoted at present. The adversarial example generation technology itself faces the problems of full convergence and easy defense. Although adversarial example technology is aggressive, it is difficult to adapt and operate in practical applications.
Response 3:
Finally, as some experts gave similar opinions on the modification of the first two sections, the homework was also revised in the original paper. However, different colors are not labeled according to different reviewer’ s opinions.
Since opinions of different experts may appear in the same section of the paper, it is impossible to mark the revisions of each expert with different colors. Please refer to the paper for specific revisions.
Thank you very much for your time and comments.

Reviewer 3 Report
In this manuscript, the authors proposes an edge-restricted adversarial example generation algorithm(Re-AEG) based on semantic segmentation. This work is interesting but the presentation skill must be improved and well written. The reviewer has some major concerns as follows:
1. The Section 0 and Section 2 should be included in Introduction. Moreover, the research background is not sufficient, please reorganize thoes part more reasonable.
2. In contribution, the authors should highlight the main point compared with the existing works. Currently, it is not enough.
3. For the proposed algorithm, the details are not clearly enough. Please further clarify this point and add the flow map or pseudo code.
4. In Simulation, the authors need to give the reason to choose the benchmark scheme. Does the other advanced algorithm can be compared in the recent works?
5. More insight should be discussed in Simulation results.
Author Response
Dear Reviewers,
Thanks very much for taking your time to review this manuscript. I really appreciate all your comments and suggestions.We have tried our best to improve and made some changes in the manuscript.Revision notes, point-to-point, are given as follows:
Point 1:
- The Section 0 and Section 2 should be included in Introduction. Moreover, the research background is not sufficient, please reorganize thoes part more reasonable.
Response 1:
(1) according to the opinion of the reviewer, we incorporate motivations and introductions to the introduction. Accordingly background information and the logic are modified;
(2) In the Introductions section, based on different application scenarios, we introduce many papers and analyze problems existing in the current research. Please refer to the introduction part of the revised manuscript.
(3) Other reviewers also gave similar opinions on the modification of the first two sections, the authors are also revised in the original manuscript. However, different colors are not labeled according to different reviewer’s opinions.
Point 2:
In contribution, the authors should highlight the main point compared with the existing works. Currently, it is not enough.
Response 2:
The authors reorganizes the main contributions from the following aspects. First, it is emphasized that different from the traditional algorithm, the proposed algorithm is optimized based on the local area of the image to hide the local object.
Second, based on the unified adversarial framework, two different adversarial examples, (adversarial perturbation and adversarial patch), are generated and verified by experiments.
Third, the algorithm can attack one or more objects so that they cannot be detected by the detector. A successful attack can be achieved even if only part of the object is exposed.
Fourth, the algorithm is verified on two public datasets.
Point 3:
For the proposed algorithm, the details are not clearly enough. Please further clarify this point and add the flow map or pseudo code.
Response 3:
Figure 3 is revised to be the flow map of the algorithm in this paper, which introduces the overall process of our algorithm. The adversarial example generation algorithm proposed in this paper uses multiple loss functions to continuously optimize and iterate the image, and finally forms a new adversarial example image with attack characteristics. Therefore, the flow map method is more intuitive than the pseudo code method. Authors supplemented the description of the flow chart as suggested by reviewer. Please refere to figure 3 of the manuscript for details.
Point 4:
In Simulation, the authors need to give the reason to choose the benchmark scheme. Does the other advanced algorithm can be compared in the recent works?
Response 4:
- Selection of attack model
Vgg16/19 is a widely used deep neural network, which is more mature than AlexNet. VGGNet successfully constructed convolutional neural networks with 16 to 19 layers deep. Compared with the previous state-of-art network structure, the error rate of VGGNet is greatly reduced. VGG19 is deeper than VGG16, which means that attacking deeper networks may require more resources. These two models are selected to verify the performance of the algorithm when attacking networks with similar structures and different depths.
The Xception architecture comes from Inception and the residual network. They divided the channel into several channels with different receptive field sizes to obtain different receptive fields. Compared with VGGNet, it has stronger feature extraction ability.
InceptionV3 model is the third generation model in Google Inception series, which increases the feature extraction ability of the network by using convolution kernels of different sizes.
Summary:
The aforementioned AlexNet, VGGNet network, is accumulated layer by layer through convolution. Xception has the features of ResNet; InceptionV3 uses convolution kernels with different sizes to achieve the fusion of features at different scales. In this paper, we conduct experiments on three typical representative network architectures. If good results can be obtained, the effectiveness of the proposed algorithm is proved.
- Experiment of complex application scenarios
In this paper, the effectiveness of the proposed attack algorithm is proved by the following experiments:
Attack object in an image containing only a single object : success
Attack a single object in an image with two targets: success
Attack a single object in an image with multiple objects (partially masked) : success
- Comparison of existing algorithms
This paper tests the attack effect of the proposed algorithm and other three classical attack methods under the four network model architectures mentioned above. These three methods are: FGSM,BIM,ILLC. These methods already perform well in the domain and are often used as baseline methods。
Among them, FGSM is a typical attack algorithm based on optimization. BIM improves the attack effect through iteration on the basis of FGSM. ILLC in turn reduces the modified perturbation based on the BIM algorithm.
The algorithms proposed in this paper are all based on gradient optimization in principle, and belong to the same category as the aforementioned several methods.
However, considering the balance between availability, convenience, scalability and success rate in practical applications, this algorithm adds more restrictions on the location and constraints of image modification. Therefore, the comparison of these classical algorithms can reflect the advantages of the proposed algorithm in terms of performance, efficiency and success rate. For detailed experimental data, please refer to Chapter IV.

Round 2
Reviewer 1 Report
The paper has been revised well.
Reviewer 3 Report
The comments had been addressed. Please consider to accept the current version.